# Tau Transfer via Extracellular Vesicles Disturbs the Astrocytic Mitochondrial System

**DOI:** 10.3390/cells12070985

**Published:** 2023-03-23

**Authors:** Romain Perbet, Valentin Zufferey, Elodie Leroux, Enea Parietti, Jeanne Espourteille, Lucas Culebras, Sylvain Perriot, Renaud Du Pasquier, Séverine Bégard, Vincent Deramecourt, Nicole Déglon, Nicolas Toni, Luc Buée, Morvane Colin, Kevin Richetin

**Affiliations:** 1Univ. Lille, Inserm, CHU Lille, LilNCog—Lille Neuroscience & Cognition, 59000 Lille, France; 2Department of Psychiatry, Center for Psychiatric Neurosciences, Lausanne University Hospital (CHUV) and University of Lausanne, 1011 Lausanne, Switzerland; 3Laboratory of Neuroimmunology, Neuroscience Research Centre, Department of Clinical Neurosciences, CHUV, 1011 Lausanne, Switzerland; 4Lausanne University Hospital (CHUV) and University of Lausanne, Neuroscience Research Center (CRN), Laboratory of Neurotherapies and Neuromodulation, 1011 Lausanne, Switzerland

**Keywords:** tauopathies, tau spreading, extracellular vesicles, astrocytes, mitochondria

## Abstract

Tauopathies are neurodegenerative disorders involving the accumulation of tau isoforms in cell subpopulations such as astrocytes. The origins of the 3R and 4R isoforms of tau that accumulate in astrocytes remain unclear. Extracellular vesicles (EVs) were isolated from primary neurons overexpressing 1N3R or 1N4R tau or from human brain extracts (progressive supranuclear palsy or Pick disease patients or controls) and characterized (electron microscopy, nanoparticle tracking analysis (NTA), proteomics). After the isolated EVs were added to primary astrocytes or human iPSC-derived astrocytes, tau transfer and mitochondrial system function were evaluated (ELISA, immunofluorescence, MitoTracker staining). We demonstrated that neurons in which 3R or 4R tau accumulated had the capacity to transfer tau to astrocytes and that EVs were essential for the propagation of both isoforms of tau. Treatment with tau-containing EVs disrupted the astrocytic mitochondrial system, altering mitochondrial morphology, dynamics, and redox state. Although similar levels of 3R and 4R tau were transferred, 3R tau-containing EVs were significantly more damaging to astrocytes than 4R tau-containing EVs. Moreover, EVs isolated from the brain fluid of patients with different tauopathies affected mitochondrial function in astrocytes derived from human iPSCs. Our data indicate that tau pathology spreads to surrounding astrocytes via EVs-mediated transfer and modifies their function.

## 1. Introduction

Tauopathies are a group of more than 20 diseases that include Alzheimer’s disease (AD), progressive supranuclear palsy (PSP), Pick disease (PiD), frontotemporal lobar degeneration and primary age-related tauopathy.

Although all tauopathies are associated with the accumulation of tau (predominantly in neurons), there are notable differences among them in terms of (1) the affected region, (2) the presence or absence of tau inclusions in glia, and (3) the tau isoform (3R/4R) that constitutes the observed inclusions/aggregates. Indeed, as a result of alternative splicing, six major isoforms of tau coexist in the human brain, each with a microtubule-binding region consisting of three (3R tau isoforms) or four (4R tau isoforms) repeated sequences [1,2]. In the healthy human brain, the 3R and 4R tau isoforms are present at equally low levels in glial cells [3,4,5]. However, analysis of patients’ brains has revealed that some tauopathies are exclusively associated with 3R or 4R tau inclusions, whereas others are associated with mixed tau inclusions [6]. Additionally, many studies have shown the presence of tau inclusions in glial cells, including astrocytes, oligodendrocytes, and microglia [7,8,9,10].

Furthermore, there is evidence of prion-like propagation of tau pathology, especially in AD [11]. This process involves neurons and affects glial cells [12]. While the origin of tau in these cells is not yet well defined, astrocytes are a particularly interesting possibility due to their function in the tripartite synapse [13]. Tau uptake and its role in the spread of tau pathology through astrocytes have become the focus of increasing attention [14]. De Gérando and collaborators showed that astrocytic tau pathology can emerge secondary to neuronal pathology [15]. It also appears that astrocytes can internalize tau fibrils, which are subsequently degraded by lysosomes, potentially contributing to reduced spread of tau [16]. The details of these tau species and how they are shuttled from neurons to astrocytes remain currently unknown.

Considering the importance of (1) tau propagation in interconnected regions and (2) the role of astrocytes in synaptic function, it is important to study the role of astrocytes in tau propagation and the underlying mechanisms involved. A few studies have investigated tau transfer between neurons and astrocytes in cell and animal models [15,16], but the cellular mechanisms underlying this exchange of material have not been explored, and no data from human samples are available. Cells exchange material with their neighbors in various ways, and these intercellular communications are fundamental to maintaining tissue homeostasis, which is often dysregulated in patients with disease. Extracellular vesicles (EVs) have emerged as critical cell-to-cell communication regulators under physiological conditions and in disorders such as neurodegenerative diseases [17,18]. They are secreted by cells through unconventional protein secretion [19] and shuttle many components, such as nucleic acids, lipids, active metabolites, and cytosolic and cell surface proteins [20]. Due to their membrane composition, they can act as unique intercellular delivery vehicles for the transfer of pathological species between cells, thereby allowing the propagation of tau pathology.

We and others recently demonstrated that EVs isolated from AD brain-derived fluids (BDFs) play a role in the spread of human tau [21,22]. Although tau has been detected in exosomes isolated from primary astrocyte cultures [23], their involvement in tau propagation is still debated. Other researchers using in animal models have suggested that tau is mainly found in exosomes derived from microglia [9]. Whether tau accumulation in astrocytes is beneficial or deleterious to the astrocytes remains also to be confirmed. We recently observed complex topographic patterns of tau isoforms and accumulation of amyloid-β in the hippocampi of AD patients. We showed that the accumulation of 3R tau (but not phospho-4R tau) in astrocytes is exacerbated by amyloid-β accumulation and impairs mitochondrial function and ATP production, thus inducing a reduction in the number of inhibitory neurons in the hippocampus, which is associated with memory decline [24]. However, the mechanism of 3R and 4R tau accumulation and their origin in hippocampal astrocytes remain poorly understood. We considered whether EV-mediated transfer of 3R/4R tau might underlie tau accumulation in astrocytes and consequent astrocyte dysfunction. In the present work, we demonstrate that (1) 3R and 4R tau are transferred from neurons to astrocytes; (2) neuronal tau is mainly secreted in the free form, while tau isoforms are shuttled from neurons to astrocytes mainly through EVs (large EVs); and (3) EV-mediated tau accumulation, especially 3R tau accumulation, in astrocytes alters mitochondrial function.

## 2. Materials and Methods

### 2.1. Human Samples

Prefrontal brain extracts from non-demented control subjects (*n* = 5), PSP (*n* = 5), and PiD (*n* = 5) patients were obtained from the Lille Neurobank (fulfilling French legal requirements concerning biological resources and declared to the competent authority under number DC-2008-642); donor consent and ethics committee approval were obtained, and the data were protected. The samples were managed by the CRB/CIC1403 Biobank, BB-0033-00030. The demographic data are listed in Table 1.

### 2.2. Rat Primary Neuron and Astrocyte Cultures

Primary hippocampal neurons were prepared from 17-day-old Wistar rat embryos, as previously described [27]. Five days later, the cells were infected with a lentiviral vector (LV) encoding human 1N4R-V5 or 1N3R-V5 (PGK-4R-V5-SIN, PGK-3R-V5-SIN), as previously described [28]. The LVs were produced and validated according to established protocol [29,30].

Primary hippocampal astrocytes were isolated from postnatal day 1 rat pups (Wistar rats from Janvier Laboratory), as previously described [31]. Briefly, the brains were removed aseptically from the skulls, the meninges were excised under a dissecting microscope, and the hippocampus was dissected. The cells were dissociated by passage through needles of decreasing gauge (1.1 × 40; 0.8 × 40; and 0.5 × 16) 4 or 5 times with a 5-mL syringe. The cells were plated at a density of 20,000 cells per cm^2^ in 6-well plates in DMEM containing 25 mM glucose and supplemented with 10% fetal calf serum, 44 mM NaHCO_3_, and 10 mL/L antibiotic/antimycotic solution (pH 7.2) in a final volume of 3 mL/well, and incubated at 37 °C in an atmosphere containing 5% CO_2_/95% air. The culture medium was replaced 4–5 days after plating and every 2–3 days thereafter, after gently tapping the plates to remove the less adherent cells (oligodendrocytes and microglia). For the neuron-to-astrocyte tau transfer experiment, 10^5^ astrocytes (1 well) were treated with 10 µL of neuron-derived small EVs (ND-SEVs), neuron-derived large EVs (ND-LEVs), and neuron-derived free protein (ND-FFP) (obtained from 10^6^ neurons, 1 well) at 10 days in vitro (D.I.V. 10).

For the microfluidic experiment, astrocytes plated in 6-well plates were detached with trypsin at D.I.V. 10 and plated in the axonal compartment. To monitor the effects of tau accumulation in astrocytes on astrocytic mitochondria, cells were infected with an LV-encoding MitoTimer (LV-G1-MitoTimer-miR124T) at D.I.V. 7–8, as previously described [28].

### 2.3. Human iPSC-Derived Astrocyte Culture

Human iPSCs were differentiated into astrocytes, as previously described in detail [32]. Briefly, human iPSCs were plated onto poly-L-ornithine/laminin (PO/L)-coated plates and cultured in neural induction medium (DMEM/F-12 supplemented with N2 supplement (1×), B27 supplement without vitamin A (1×), noggin (500 ng/mL), SB431542 (20 μM), and FGF2 (4 ng/mL)) to allow the formation of embryoid bodies. The medium was changed every other day for 12 days. After 12 days, neural precursor cells (NPCs) were passaged and grown for at least 3 weeks in single-cell culture in NPC expansion medium (DMEM/F-12 supplemented with N2 supplement (1×), B27 supplement without vitamin A (1×), FGF2 (10 ng/mL), and EGF (10 ng/mL)). Then, the medium was replaced with astrocyte induction medium (DMEM/F-12 supplemented with N2 supplement (1×), B27 supplement without vitamin A (1×), LIF (10 ng/mL), and EGF (10 ng/mL)). The medium was changed every other day, and the cells were passaged when they reached confluence. After 2 weeks, the medium was replaced with astrocyte medium (DMEM/F-12 supplemented with B27 supplement without vitamin A) supplemented with CNTF (20 ng/mL). After culture for 4 weeks in this medium, the cells displayed the cellular characteristics of mature human astrocytes [33]. Then, the astrocytes were infected with an LV-encoding MitoTimer (LV-G1-MitoTimer-miR124T). Four days later, the astrocytes were treated with 10 µL of BDF-LEVs (10^6^ EVs per astrocyte), and mitochondrial system function and dynamics were evaluated. The dose of EVs was selected based on a cell toxicity experiment (the number of DAPI-stained cells remaining 24 h after treatment was counted and normalized with the control); in this experiment, no toxicity was observed at this dose (Appendix A). These experiments were performed according to a protocol approved by our institutional review committee (approval n° CER-VD 2018-01622).

### 2.4. Microfluidic System and Immunofluorescence

Glass coverslips were coated overnight at 4 °C with 0.5 mg/mL poly-D-lysine. Microfluidic chambers (AXIS™, Temecula, CA, USA) were subsequently placed on the coated glass coverslips and attached to the glass. All chambers had microgrooves 450 μm in length and 10 μm in width. Rat primary embryonic neurons were cultured as described above, and approximately 30,000 cells were plated in the two wells of the somatodendritic compartment. The cultures were maintained at 37 °C for 7 days to allow differentiation, with a volume gradient from the somatodendritic compartment to the astrocytic compartment to assist with axonal guidance. At D.I.V. 7, Tau-V5-LV (200 ng of LV per well) was added to the somatodendritic compartment after first reversing the volume gradient between the compartments to counteract viral diffusion. The quality controls for compartment isolation were as previously described by Dujardin and collaborators [28]. Primary astrocytes were then plated in the axonal compartment in neurobasal medium and maintained at 37 °C for 5 days. V5 immunolabeling was performed to evaluate tau transfer from neurons to astrocytes. The cells in the compartments (somatodendritic and axonal) were washed once with phosphate-buffered saline (PBS) and fixed with 4% paraformaldehyde (PFA) for 20 min. After removing the fixative, the cells were washed three more times with 50 mM NH4Cl and permeabilized with Triton X-100 (0.1%, 10 min at room temperature). Subsequently, the slides were incubated with primary antibodies at 4 °C overnight, and labeling was performed by incubation with the appropriate Alexa Fluor conjugated secondary antibodies (1:400) for 45 min at room temperature. The cells were mounted with Vectashield medium containing DAPI.

### 2.5. EVs Isolation from Primary Cultures

Neuron media (10 × 10^6^; 10 wells) were collected and placed on ice 10 days after LV infection. Protease inhibitors were added before centrifugation for 10 min at 2000× *g* and 4 °C. The supernatant was centrifuged for 50 min at 20,000× *g*, and the pellet, which contained ND-LEVs, was collected. The supernatant was then centrifuged for 50 min at 100,000× *g* to obtain ND-SEVs (pellet) and FFP (supernatant). ND-LEVs and ND-SEVs were suspended in 100 µL 100 µL of 4% PFA (diluted in phosphate buffer (0.08 M Na_2_HPO_4_ and 0.02 M NaH_2_PO_4_)) for electron microscopy analyses or 100 µL of phosphate buffer (0.08 M Na_2_HPO_4_ and 0.02 M NaH_2_PO_4_) for astrocyte treatment. The FFP was concentrated to a volume of 100 µL using an Amicon device (3 kDa).

### 2.6. EV Isolation from Human Prefrontal Cortex Tissue

BDF-EVs were isolated from human prefrontal cortex tissue as previously described [22,34]. The fluid was a suspension obtained by gentle papain digestion of brain extract and not a directly collected fluid sample. Briefly, the tissue was incubated on ice in 5 mL of Hibernate-A. It was gently mixed in a Potter homogenizer, and 2 mL of 20 units/mL papain in Hibernate-A was added to the homogenate for 20 min at 37 °C with agitation. Then, 15 mL of cold Hibernate-A buffer (50 mM NaF, 200 nM Na_3_VO_4_, 10 nM protease inhibitor) was added, and the sample was mixed by pipetting to stop the enzymatic activity. Successive centrifugation was performed at 4 °C (300, 2000, and 10,000× *g*) to remove cells, membranes, and debris, respectively. The final supernatant was kept at −80 °C until EV isolation was performed. BDF-LEVs and BDF-SEVs were pooled from 5 patients per group. Isolation of EVs from cell medium or human BDF was carried out using differential ultracentrifugation as previously described [35] to obtain LEV-, SEV- and FFP-enriched fractions. The final pellets were suspended in a final volume of 400 µL and kept at −80 °C.

### 2.7. Nanoparticle Tracking Analysis (NTA)

The size, number, and distribution of particles were determined using NTA (Nanosight NS300, Malvern, Cambridge, United Kingdom) as described previously [22]. To generate statistical data, five 90-s videos were recorded and analyzed using NTA software (camera level: 15; detection threshold: 4).

### 2.8. Electron Microscopy [22,35]

Samples (5 μL) were deposited on a carbon film-supported grid (400 mesh) and incubated at room temperature (RT) for 20 min. The grids were fixed in PBS-glutaraldehyde (1%) for 5 min at RT and then rinsed in distilled water. They were incubated for 5 min in 1% uranyl acetate and then for 10 min on ice in a mixture of 1% uranyl acetate/2% methylcellulose. Dry grids were observed under a transmission electron microscope (Zeiss EM900). When indicated, immunolabeling was performed. The grids were rinsed once in PBS and incubated twice (3 min at RT) in PBS-50 mM glycine before incubation in PBS-1% bovine serum albumin (BSA) for 10 min at RT. A primary antibody diluted in PBS-1% BSA was applied for 1 h at RT and detected using an appropriate secondary antibody diluted in PBS-1% BSA (18 nm gold colloidal goat anti-mouse). After rinsing in PBS, the grids were processed as described above.

### 2.9. Antibodies

The following antibodies were used for immunohistofluorescence and immunohistochemistry at the indicated dilutions: a polyclonal rabbit antibody against the C-terminus of Tau (C-Ter), which recognizes the last 15 AA of the protein (C-Ter, raised in house, 1:1000 for electronic microscopy) [36]; a mouse monoclonal antibody against V5 (1:10,000) that recognizes the V5 epitope of tagged tau [24,28]; and a polyclonal rabbit antibody that recognizes glial fibrillary acidic protein (GFAP) (1:10,000).

### 2.10. ELISA

Fractions were obtained after ultracentrifugation of culture medium or BDF as described above, and 10 µL were added to astrocytes. Tau levels were then determined using INNOTEST hTau Ag (Fujirebio/Innogenetics, Gent, Belgium), which is a sandwich ELISA microplate assay for the quantitative determination of human tau antigen levels in fluids, according to the manufacturer’s instructions. The capture antibody was an AT120 antibody, and biotinylated HT7 and BT2 antibodies were used as detection antibodies [37]. For cell samples, we customized the INNOTEST hTau Ag ELISA kit to quantify Tau-V5 levels. The original INNOTEST kit plate was replaced with a 96-well MicroWell™ MaxiSorp™ flat bottom plate coated with V5 antibody (Invitrogen P/N-0705; 1:10,000) in carbonate buffer (NaHCO_3_ 0.1 mM, Na_2_CO_3_ 0.1 mM; pH = 9.6) overnight at 4 °C. The other steps were performed according to the manufacturer’s instructions. Tau uptake by astrocytes was evaluated 24 h after incubation of astrocytes with ND-FFP and ND-LEV fractions from control neurons (NI) or neurons overexpressing 1N3R or 1N4R Tau-V5. Tau uptake efficiency was calculated with the following equation: % of tau uptake= [Quantity of tau in astrocytes x 100]/[Quantity of tau added to astrocytes].

### 2.11. Quantification of Tau-V5 Levels in Primary Culture

The optical density (OD) of Tau-V5 was measured in 5 regions of interest (ROI) in each compartment (somatodendritic compartment, microgrooves, and axonal compartment). The optical density in each region of interest was evaluated using Zen 2 image analysis software (blue edition). Each OD value was normalized by subtracting the OD in an ROI in which no Tau-V5 signal was present (NI). The OD of Tau-V5 in astrocytes was measured from confocal scanning images taken with a Zeiss LSM 710 Quasar microscope equipped with a 40× oil immersion objective at a *z*-axis resolution of 0.9 µm. We used Imaris software to measure the intensity of Tau-V5 in the reconstructed GFAP+ region of the axonal compartment.

### 2.12. Analysis of Mitochondrial Redox State, Morphology and Dynamics by MitoTimer

We recently developed a new lentiviral vector (LV-G1-MitoTimer-MiR124T, hereafter called LV-G1-MitoTimer) to study the dynamics and functions of mitochondria, specifically in astrocytes in vitro and in vivo. LV-G1-MitoTimer uses a truncated version of the glial fibrillary acidic protein (GFAP) promoter gfaABC1D, with a B3 enhancer (gfaABC1D(B3), hereafter called G1) combined with the previously described miR124T neuronal detargeting system. It allows exclusive expression of the mitochondrial biosensor in astrocytes in vitro and in vivo. MitoTimer is a mutant red fluorescent protein, drFP58317, with a mitochondrial signal from subunit VIII of human cytochrome c oxidase, capable of visualizing newly synthesized mitochondria in green (488 nm) and oxidized mitochondria in red (555 nm). The green (488 nm) and red (555 nm) fluorescence ratio enables simultaneous evaluation of individual mitochondria, their morphology analysis, fusion/fission events, and redox state history. This unique property can be applied to investigate many scientific questions regarding mitochondria’s physiological and pathological roles, and is therefore very promising for unveiling the underlying mechanisms of mitochondrial dynamics within many different cell types [38,39]. Briefly, using an inverted microscope (Nikon Eclipse Ti-2, Melville, NY, USA) (150× magnification, 100× oil immersion objective, 1.5× intermediate magnification) and the Perfect Focus System (PFS), 16 bit image sequences (1 frame/s for 60 s) were taken at baseline (BL) and 6 h and 24 h after treatment with EVs. Sequential excitation at 490 nm (for the green channel) and 550 nm (for the red channel) and detection of green (500–540 nm) and red (550–600 nm) signals were carried out. Then, using GA3 software, we selected the first frame of each image sequence in the red and green channels to generate binary masks for each mitochondrion. The resulting segmentations were filtered by intensity and size to remove poor-quality objects, and by position within the region of interest (ROI) to remove objects truncated by the ROI boundary. We extracted statistics of individual mitochondria (masks) such as elongation factor (length to width) and mitochondrial length, as well as the ratio of the average red (555 nm) to green (488 nm) intensity corresponding to the redoxstate ratio. We performed these analyses for at least 20–25 cells per condition (a minimum of fifty mitochondria per cell). Then, we considered mitochondria that could be tracked along the entire sequence length to analyze the mitochondrial dynamics. We extracted the following parameters for each reconstructed track: the total displacement (displacement) corresponding to the distance (um) between the track’s beginning and end and the average speed (speed) in um/s. Analysis of the fusion and fission events was performed manually on identical images with the successive counts of objects normalized to the average number of mitochondria per image constructed from the previous images. This resulted in the estimation of the event rate per particle. The event rate can provide information on the activity of fusion and fission events, among the most critical mechanisms of mitochondrial dynamics. In addition, we calculated the number of branches for each reconstructed mitochondria, indicating the tendency of mitochondria to assemble and form complex structures observed in astrocytes. Nikon’s NIS Element system was employed to manually track 25–50 mitochondria per cell. Finally, after log transformation, the change in score [39] relative to the BL was calculated and normalized to that of the control group (ND-LEVs-CFP for primary cultures and BDF-LEVs from human controls) [38].

### 2.13. Sample Sizes, Calculations and Statistical Analysis

Human samples were classified based on neurological and neuropathological examination. The order of culture and procedures was randomized for each experiment. Investigators were blinded to group allocation when processing the tissue and performing cell counts and during confocal image acquisition. Values are presented as the mean  ±  s.e.m.; N corresponds to the number of independent experiments, and n corresponds to the overall number of values. Statistical analyses of raw data were performed with GraphPad Prism software v8.0. The normality of the data was verified using the Shapiro-Wilk test. Differences between two experimental groups were analyzed using Student’s *t* test (parametric test) or the Mann–Whitney test (nonparametric test). Statistical analyses of differences among more than two experimental groups were performed using one-way ANOVA followed by Dunn’s post hoc analyses for multiple comparisons (parametric test) or the Kruskal–Wallis test (nonparametric test). For MitoTimer score change analysis, as previously described [38], we used two-way ANOVA followed by Tukey’s post hoc analysis with time as the independent variable to compare the differences in mitochondrial function between the 3R and 4R tau groups. In addition, we used the Wilcoxon signed-rank test to compare each parameter measured at 6 h and 24 h with the baseline values for each astrocyte.

## 3. Results

### 3.1. 3R and 4R Tau Transfer from Neurons to Astrocytes

To study the origin of the accumulated 3R and 4R tau in astrocytes, we overexpressed human wild-type 1N3R and 1N4R tau (fused to a V5 tag) in primary hippocampal neuron cultures in a microfluidic system. Due to axonal growth in the microgrooves, the hippocampal neurons in the somatodendritic compartment (SD) were connected to a second chamber (axonal compartment, AX) containing primary astrocytes. 1N3R or 1N4R tau was overexpressed in the somatodendritic compartment using LVs (Figure 1A). Five days later (D.I.V. 12), immunochemistry for V5 revealed accumulation of 1N3R and 1N4R tau in the somatodendritic (containing soma and dendrites) compartment and axons in the microgrooves (Figure 1B). Both isoforms were overexpressed in neurons at similar levels (Figure 1C). Interestingly, we observed that a large population of astrocytes in the axonal compartment (identified by GFAP immunochemistry) exhibited Tau-V5 puncta in the cytoplasm (Figure 1B,D). Quantification of 1N3R and 1N4R tau inclusions in astrocytes through measurement of the V5 intensity in GFAP+ astrocytes indicated that both isoforms could be transferred from hippocampal neurons to astrocytes in similar degrees (Figure 1E).

### 3.2. Neuronal Tau Is Mainly Secreted in the Free Form

Transfer of tau to adjacent cells requires secretion by the donor cell. First, we assessed which form(s) of tau is (are) secreted by primary neurons. Culture media from primary neurons overexpressing 1N3R or 1N4R tau at similar levels (Figure 2A,B) were collected and fractioned to separate EVs from free proteins. EVs present in extracellular media are a heterogeneous population in which different biogenesis pathways are active. EVs include (1) exosomes (small vesicles, size < 150 nm), which are generated from multivesicular bodies containing intraluminal vesicles that are secreted into the extracellular fluid, and (2) ectosomes (large vesicles, size > 150 nm) that originate directly from budding at the plasma membrane [20]. To further investigate the EV subtypes that might be involved in tau transfer to astrocytes, we separated small from large vesicles as previously reported [35]. Fractions containing ND-LEVs, ND-SEVs, and ND-FFP were then collected and characterized. Electron microscopy (Figure 2C) and NTA (Figure 2D,E) confirmed that ND-SEVs and ND-LEVs with intact structures were present in our fractions. Totals of 26.3 ± 5.8% and 35 ± 18.7% ND-SEVs of size > 150 nm and 73.6 ± 5.8% and 65 ± 18.7% ND-SEVs of size between 10 and 150 nm were isolated from cells overexpressing 1N3R and 1N4R tau, respectively (Figure 2D). Totals of 71.3 ± 9.2% and 63.6 ± 13.8% ND-LEVs of size > 150 nm and 28.6 ± 9.2% and 36.3 ± 13.8% ND-LEVs of size between 10 and 150 nm were isolated from cells overexpressing 1N3R and 1N4R tau, respectively (Figure 2E). It should be noted that size repartition was not altered by the expression of different tau isoforms in neurons (Figure 2D,E nonsignificant difference between the 1N3R and 1N4R groups). The presence of tau in these fractions was detected by electron microscopy (Figure 2F), and tau levels were quantified by ELISA (Figure 2G). While a large majority (93% ± 0.2 and 93.3 ± 0.2 for 1N3R and 1N4R tau, respectively) of tau protein was found in the free form (ND-FFP), only 4.7% ± 0.3 of 1N3R tau and 4.5% ± 0.2 of 1N4R tau was found in the ND-LEV fractions (Figure 2G).

### 3.3. Tau Isoforms Are Shuttled from Neurons to Astrocytes Mainly by EVs

The three fractions were applied to rat hippocampal astrocyte cultures, and tau transfer was evaluated at 24 h (homemade ELISA Tau-V5 and immunofluorescence) (Figure 3A). The presence of 1N3R and 1N4R tau in astrocytes after ND-LEV treatment was detected by immunofluorescence with antibodies against the V5 epitope and GFAP (Figure 3B), and 1N3R and 1N4R tau levels in astrocytes were quantified by ELISA (Figure 3C). Although the vast majority of tau secreted by neurons was in the free form (Figure 2G), only the human 3R isoform was detected after treatment of astrocytes with ND-FFP. In contrast, among EVs, only ND-LEVs had the ability to deliver both human 1N3R and 1N4R tau to astrocytes (Figure 3C). We then considered whether tau in ND-LEVs is transferred from neurons to astrocytes more efficiently than in ND-FFP. We normalized the [Tau-V5] in astrocytes to the [Tau-V5] in the fractions applied to astrocytes and secreted by neurons (ND-FFP and ND-LEVs). The percentage of tau taken up by astrocytes was significantly different between the ND-FFP and ND-LEV groups, and tau accumulation was not significantly affected by the tau isoform (Figure 3D). Altogether, these results highlight the major role of LEVs in tau transfer between neurons and astrocytes.

### 3.4. 3R and 4R Tau-Accumulating LEVs Treatments Induce Differential Mitochondrial Consequences in Astrocytes

We previously demonstrated that the accumulation of different human tau isoforms in astrocytes differentially affects the astrocytic mitochondrial system [24]. The effect of ND-LEVs derived from neurons accumulating 3R or 4R tau was assessed by following a MitoTimer biosensor with high-content live microscopy (Figure 4A). This multiparametric approach can reveal changes in individual mitochondria over several hours/days [38]. Mitochondrial features (including the redox state, morphology, and dynamic changes) were measured before (BL) ND-LEV^CFP^, ND-LEV^3R^, or ND-LEV^4R^ treatment and at 6 h and 24 h after treatment. For each astrocyte- and each mitochondria-related criterion, the change score was calculated relative to the BL value and normalized to that of the control group (ND-LEVs^CFP^).

Six hours after ND-LEV^3R^ or ND-LEV^4R^ treatments, we observed that the number of mitochondrial events (fusion/fission) significantly increased compared with the control condition ND-LEVs^CFP^ (Figure 4B). However, only ND-LEVs^4R^ increased the morphological complexity by significantly increasing the number of branches of mitochondria (Figure 4B). At 6H, multiple *t*-tests between treatments indicated that the mitochondrial branch criteria distinguish between the two types of ND-LEV^3R^ or ND-LEV^4R^ treatments (Figure 4B). Twenty-four hours after ND-LEV^3R^ or ND-LEV^4R^ treatments, we observed that ND-LEVs^3R^ significantly increased the redox state of mitochondria and reduced the number of events and the elongation of astrocyte mitochondria (Figure 4C). Conversely, ND-LEVs^4R^ reduced the redox state and tended toward more branched mitochondria. Multiple *t* tests between treatments revealed that the mitochondrial redox state and elongation significantly differed between ND-LEV^3R^ and ND-LEV^4R^ treatments (Figure 4C). These data suggest that the mitochondrial system of astrocytes is rapidly sensitive (6h) to the pathological content of LEVs, and the effect may be damaging in the longer term (24H).

Then, we evaluated changes in mitochondrial parameters (24 h) in human iPSC-derived astrocytes (expressing the biosensor MitoTimer) treated with BDF-LEVs isolated from a pool of patients diagnosed with 3R-tau (PiD) or 4R-tau (PSP) tauopathy, and non-demented control subjects (C) (Figure 5A and Appendix A). The size, concentration, and protein content of BDF-LEVs were initially measured using NTA and chromatography–tandem mass spectrometry (LC–MS/MS) (Appendix A). Regarding redox state, astrocytes treated with BDF-LEVs^PID^ exhibited significantly increased mitochondrial oxidation, whereas mitochondrial oxidation was significantly reduced in astrocytes treated with BDF-LEVs^PSP^ (Figure 5B,C). Moreover, BDF-LEVs^PID^ significantly reduced mitochondrial elongation, whereas the mitochondria of astrocytes treated with BDF-LEVs^PSP^ appeared significantly more elongated and widely distributed in the cell (Figure 5B,D). Altogether, these results revealed that EVs from cells with 3R tau accumulation had deleterious effects on mitochondrial redox state and morphology, while EVs from cells with 4R tau accumulation induced filamentation and had less severe functional effects on the astrocytic mitochondrial system.

## 4. Discussion

This study investigates the involvement of glial cells and EVs in the spread of tau. To date, most studies addressing the spreading process have focused on neurons. These cells are indeed the most affected cells in tauopathies, especially in AD, the main feature of which is the aggregation of tau protein (3R and 4R tau) into paired helical filaments within neurons. Nevertheless, tau inclusions are also found in glial cells, including astrocytes, in other primary tauopathies. Astrocytes are the principal glial cells in the brain and play a supporting role in supplying energy and nutrition to neurons. Evidence shows that astroglial atrophy occurs in early stages of neurodegeneration, potentially leading to disruption in synaptic connectivity [40]. Astrocytes form a direct link between pre- and postsynaptic neurons and may therefore be involved in the propagation of tau between interconnected brain regions. Moreover, tau in astrocytes involved in tripartite synapses can impair synaptic function, as has been described in the brains of AD patients [41].

Here, we evaluated whether tau protein is shuttled from neurons to astrocytes via EVs, seeking a potential explanation for the accumulation of tau in astrocytes in the AD brain [24]. We first investigated the related cellular mechanisms in primary neurons overexpressing either 3R or 4R tau. EVs from murine primary neurons expressing human 3R or 4R tau were isolated and applied to murine primary astrocytes. Whereas tau is mainly secreted in a free form, EVs transfer both the two isoforms to astrocytes in equal quantities. In agreement with the work of Mate de Gerando and colleagues [15], we confirmed that tau is transferred from neurons to astrocytes. Narasimham and collaborators also demonstrated that while oligodendroglial tau pathology propagated across the mouse brain in the absence of neuronal tau pathology, astrocytic tau pathology did not. This suggests that pathological forms present in astrocytes are derived from neurons, and supports the existence of tau spreading between neurons and astrocytes [42].

Most importantly, through a direct comparison of free and EVs-associated tau, we validated the high potential of EVs to shuttle tau between neurons and astrocytes. This comparison also highlights the importance of brain-derived EVs in spreading tau pathology in Alzheimer’s disease [21,22]. Contrary to our findings, some other studies have reported that free tau is taken up in astrocytes. They applied full-length recombinant proteins (2N4R isoform) [5] or preformed fibrils of human tau with P301L mutation not found in primary tauopathies such as PSP or Pick’s disease (K18-P301L) [16]. In our experimental design, we applied neuronal tau secretome. We speculate that this secretome is closer to pathology, as many studies now validated the presence of tau peptides in biological fluids and not exclusively the full-length tau.

We then studied the effects on astrocytes of EV-mediated 3R or 4R tau transfer. Our data demonstrated that the transfer of EVs from neurons with 3R or 4R tau accumulation induced precise changes in the astrocytic mitochondrial system. Indeed, EVs originating from neurons with 3R tau accumulation rapidly showed features of mitochondrial dysfunction/damage (a transient increase in the number of events that result in fragmentation, and a substantial rise in the mitochondrial redox state). In contrast, astrocytes treated with EVs originating from neurons with 4R tau accumulation show features of transient adaptation/compensation of the mitochondrial system (transient ramification, a reduction in the redox state and/or an increase in mitochondrial turnover) [43,44]. Similar results were obtained after overexpression of 3R and 4R tau in astrocytes [24], whereas opposite effects were observed in neurons [45].

Numerous studies have already shown that the accumulation of abnormal tau protein impairs mitochondrial function, which leads to cell degeneration. These alterations in function include aspects of mitochondrial transport, dynamics, bioenergetics, and mitophagy [46].

Because tau is a microtubule-binding protein, mitochondrial disturbances due to abnormal tau had long been attributed to changes in the organization of the cytoskeletal network [47]. However, several recent studies have demonstrated that abnormal tau has a direct effect on mitochondria via binding. Indeed, it has been shown that abnormal tau can alter MFN2 levels and trigger the mislocalization and clustering of DRP1, triggering the elongation of mitochondria [48]. Abnormal tau can also directly interfere with the Parkin protein, inhibit mitophagy [49], and significantly reduce the activity of complex I, voltage-dependent anion channels (VDACs), and respiratory complex V subunits [50]. Abnormal tau protein may also impact ER-mitochondrial coupling, which could affect all mitochondrial functions [51]. The large majority of related studies have used neurons and 4R tau isoforms. Consequently, the importance of the specific tau isoform in determining the extent of mitochondrial alterations remains unclear.

Our studies demonstrated that 3R and 4R tau are transferred from neurons to astrocytes with the same efficiency. We can hypothesize that 3R and 4R tau act on astrocytic mitochondria through different molecular mechanisms. First, whether three or four microtubule-binding domains are present slightly influences the binding affinity of tau to microtubules. Indeed, 4R tau binds to microtubules with a three-fold higher affinity than 3R tau [52]. We can also assume, as demonstrated in neurons, that tau protein may bind differently to filamentous actin to induce the formation of aligned bundles of actin filaments, therefore modifying the organization of the cytoskeletal network [47]. Second, because it is more soluble and has stronger kinesin inhibitory activity than 4R tau, 3R tau may induce strong steric inhibition of the binding of mitochondria to microtubules, leading to their immobilization [53]. Furthermore, the presence of a single cysteine (C322) in 3R tau allows the formation of intermolecular bridges, whereas the presence of two cysteines (C291, C322) in 4R tau leads mainly to the formation of intramolecular bridges [54]. These differences in the ability to form inter- or intradisulfide bridges may explain why 3R tau more readily aggregates to form oligomers and polymers than 4R tau. If these differences in properties are maintained in EVs, they could explain the differential effects of 3R and 4R tau on mitochondria observed in our study. Indeed, EVs carrying 3R tau may contain more tau peptides that are directly toxic to mitochondria [21,22,46] than EVs containing 4R tau. To extrapolate this effect to the human condition, we next isolated EVs from the brain fluids of patients with a 3R tau-related primary tauopathy (PiD) and a 4R tau-related primary tauopathy (PSP). Recently, we and others showed that EVs isolated from tauopathy patient-derived brain fluids contain tau seeds that might be involved in tau spreading [21]. Application of these EVs to iPSC-derived astrocytes validated their deleterious effect on mitochondria and showed that EVs from pure 3R tau-related tauopathy samples are the most aggressive, confirming our data from murine primary cultures. Altogether, our data are consistent with our previous study showing that 3R and 4R tau overexpression in astrocytes differentially alters the mitochondrial localization, trafficking, and function of these tau isoforms, suggesting that astrocytes may play a more substantial role than expected in AD [24] and other pure tauopathies. However, it is very important to mention that the observed effects on mitochondria may be more complex than the effect of tau alone. Indeed, EVs carry many other proteins (or protein peptides) and mRNAs that could impact the astrocytic mitochondrial system. Recent evidence suggests that the transfer of mitochondrial content by EVs modifies metabolic and inflammatory responses in recipient cells [55]. Alone or in combination, these effects are consistent with our observations of altered mitochondrial transport and dynamics after EV-mediated 3R tau transfer.

Here, we found that human-derived EVs play a role in astrocytic dysregulation. We also described a new potential pathway for the spread of tau between interconnected regions surrounding the synaptic cleft. We showed that EVs play a significant role in propagating 3R and 4R tau to astrocytes. Treatment with accumulated tau-containing EVs originating from neurons with 3R tau accumulation disturbed the astrocytic mitochondrial system and had very damaging effects. These data are in accordance with the severity of Pick disease. Indeed, in contrast to PSP, Pick disease is much more aggressive and damaging, involving huge cortical atrophy [56]. The current study raises many questions about strategies for clearing free extracellular tau. However, targeting pathological EVs is very challenging, and further studies are required to characterize EV cargos/transport, as recently described [57,58,59]. This will help not only in differentiating tauopathies but also in designing specific tools to block tau spreading.

## 5. Conclusions

In this study, we have discovered new mechanisms that explain how pathology spreads from neurons to surrounding astrocytes and alters their functioning. We utilized several innovative materials and technologies, including viral vectors encoding human tau isoforms, IPSC-derived astrocytes, human brain samples, microfluidic chambers, and live high-resolution imaging of mitochondrial biosensors. Our findings shed light on how astrocytes uptake tau and the functional consequences of this process. Specifically, we demonstrate that human tau 3R and 4R are equally transferred from neurons to astrocytes, that tau isoforms are mainly transported via LEVs from neurons to astrocytes, and that exposure to LEVs with tau 3R pathology has devastating effects on the mitochondrial system of recipient astrocytes.

## Figures and Tables

**Figure 1 cells-12-00985-f001:**
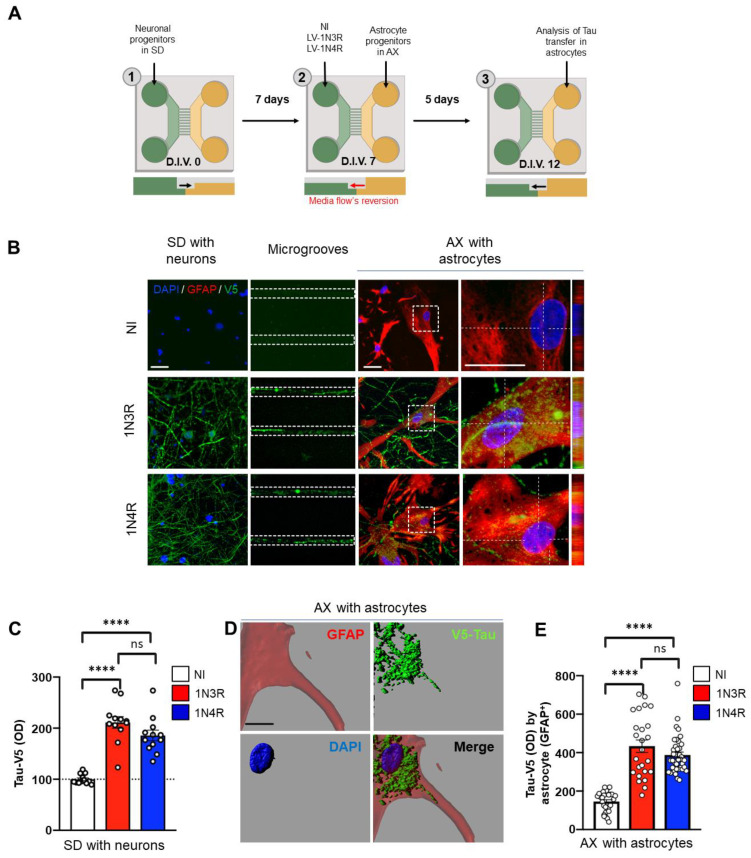
Tau is shuttled from neurons to astrocytes. (**A**) Schematic of the microfluidic system used to investigate tau transfer from hippocampal neurons to astrocytes. (**B**) Confocal micrographs showing 1N3R or 1N4R tau (V5+) and astrocytes (GFAP+) in the somatodendritic and axonal compartments and microgrooves. (**C**) Histogram showing the Tau-V5 optical density in the somatodendritic compartment. (**D**) 3D reconstruction of confocal micrographs showing tau (V5+) in an astrocyte (GFAP+) in the axonal compartment. (**E**) Histogram showing the number of Tau-V5 inclusions detected per astrocyte. The scale bars are 20 µm (**B**) and 5 µm (**D**). SD = somatodendritic compartment, AX = axonal compartment, LV = lentiviral vector, NI = noninfected. For (**C**,**E**), N  =  cultures/microfluidic chambers/cells: NI: 1/3/32, 1N3R: 1/3/25, 1N4R: 4/3/39. Ordinary one-way ANOVA with Sidak’s multiple comparison test. (^ns^
*p* >0.05, **** *p* < 0.0001).

**Figure 2 cells-12-00985-f002:**
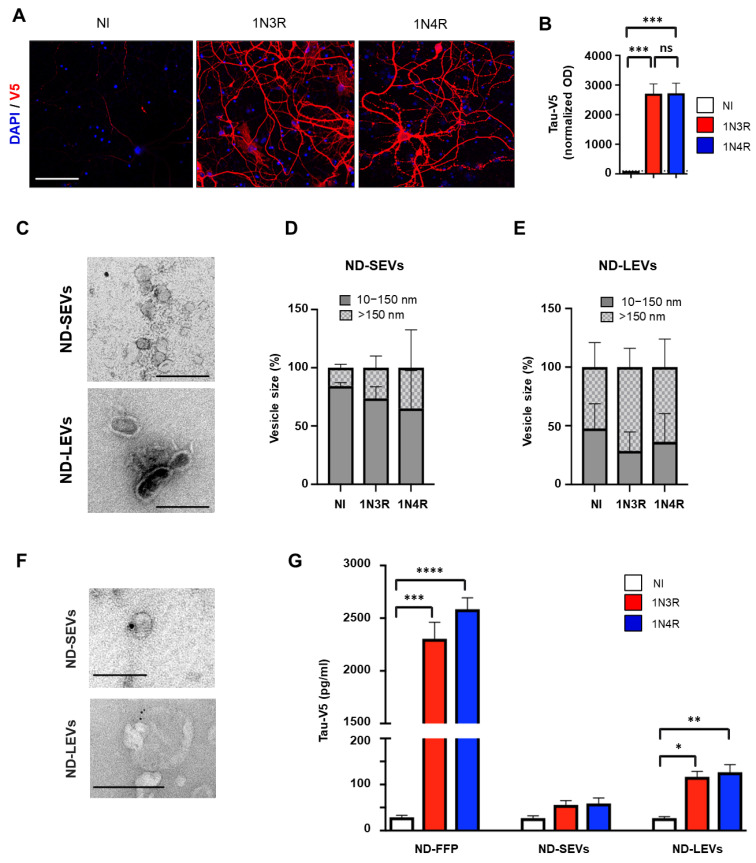
Neuronal tau is mainly secreted in a free form. (**A**) Confocal micrographs showing tau (V5+) in rat hippocampal neurons infected by LVs overexpressing 1N3R or 1N4R tau; NI = noninfected. (**B**) Histogram showing Tau-V5 accumulation (optical density normalized to that in the NI condition) in rat hippocampal neurons overexpressing 1N3R or 1N4R tau. (**C**) Electron microscopy images of ND-SEVs and ND-LEVs isolated from cultured neurons. The scale bar is 250 nm. (**D**) Histogram showing the percentage of ND-SEVs with a size between 100–150 nm and greater than 150 nm. (**E**) Histogram showing the percentage of ND-LEVs with a size between 100–150 nm and greater than 150 nm. (**F**) Electron microscopy and immunogold labeling (Cter-tau) of ND-SEVs and ND-LEVs isolated from the supernatant of neurons overexpressing 1N4R tau. The scale bar is 100 nm for ND-SEVs and 250 nm for ND-LEVs. (**G**) Histogram showing Tau-V5 concentration in the ND-FFP, ND-SEV, and ND-LEV fractions from control rat hippocampal neurons (NI) and those overexpressing 1N3R or 1N4R tau. For (**B**,**G**), *n*  =  4 cultures per condition; ordinary one-way ANOVA with Sidak’s multiple comparison test and the nonparametric Kruskal–Wallis test, in (**B**) and (**G**), respectively. (^ns^
*p* > 0.05, * *p* < 0.05, ** *p* < 0.01, *** *p* < 0.001, **** *p* < 0.0001).

**Figure 3 cells-12-00985-f003:**
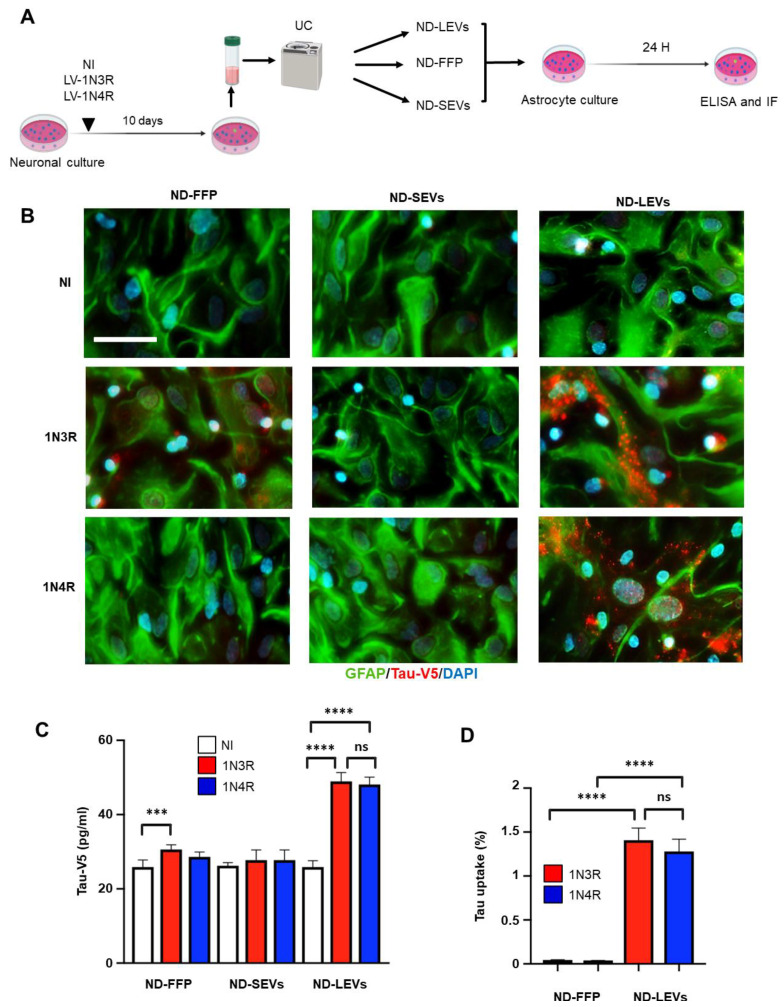
Tau is shuttled from neurons to astrocytes via LEVs. (**A**) Schematic representation of the protocol employed to isolate ND-EVs from neurons infected by LVs overexpressing 1N3R or 1N4R tau. ND-EVs from NI cultures were used as controls. (**B**) Example of confocal images showing the transfer of ND-FFP, or tau in ND-SEVs and ND-LEVs from neurons overexpressing 1N3R or 1N4R tau. The scale bar is 25 µm. (**C**) Histogram showing the Tau-V5 concentration in astrocytes 24 h after incubation with the ND-FFP, ND-SEV, and ND-LEV fractions from control rat hippocampal neurons (NI) and those overexpressing 1N3R or 1N4R tau. (**D**) Histogram showing the tau uptake efficiency 24 h after incubation with the ND-FFP and ND-LEV fractions from control rat hippocampal neurons (NI) and those overexpressing 1N3R or 1N4R tau. For (**C**) and (**D**), *n* = 4 cultures per condition; ordinary one-way ANOVA with Sidak’s multiple comparison test. (^ns^
*p* > 0.05, *** *p* < 0.001, **** *p* < 0.0001).

**Figure 4 cells-12-00985-f004:**
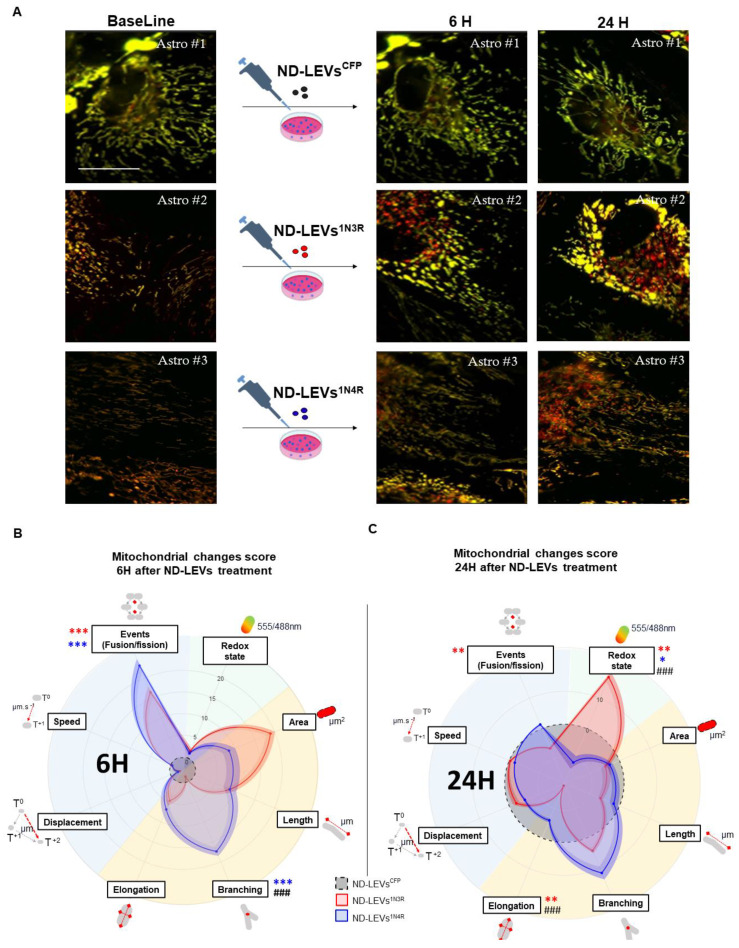
Tau isoform-containing LEVs induce mitochondrial dysfunction in primary rat astrocytes. (**A**) Micrographs of the mitochondria labeled with the biosensor MitoTimer before (BL) and after (6 h and 24 h) treatment with ND-LEVs^CFP^, ND-LEVs^1N3R^, and ND-LEVs^1N4R^. (**B**,**C**) Radar charts showing MitoTimer ratio 555/488nm, morphology (surface area, length, number of branches, factor of elongation,), mobility (displacement and speed), and number of event changes (fusion/fission), normalized to BL values and the value of the ND-LEV^CFP^ group, in astrocytes treated with ND-LEVs^1N3R^ and ND-LEVs^1N4R^. Two-way matched ANOVA followed by post hoc analysis (Tukey’s test) was applied to compare the effect of each treatment with that of ND-LEVs^CFP^ (* *p* < 0.05, ** *p* < 0.01 and *** *p* < 0.001). A multiple *t* test was performed on each criterion between ND-LEVs^1N3R^ and ND-LEVs^1N4R^ to evaluate the difference in effect between the 3R and 4R isoforms ### *p* < 0.001).

**Figure 5 cells-12-00985-f005:**
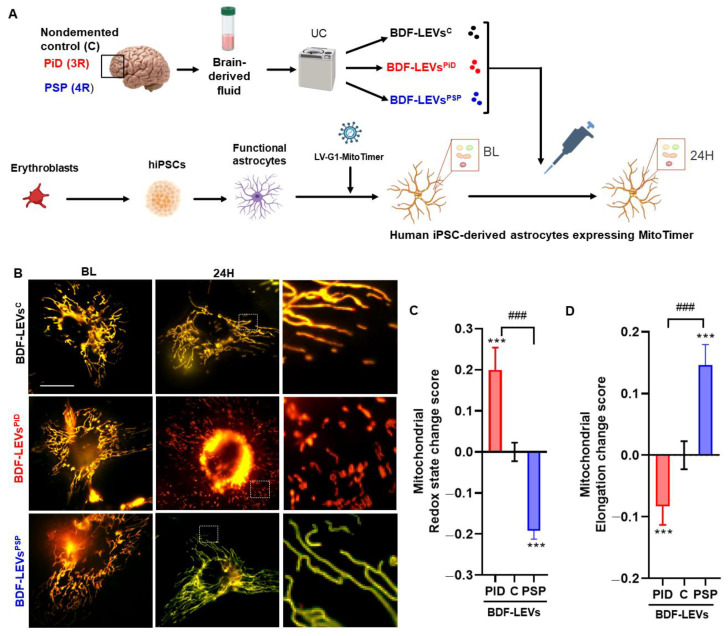
LEVs from 3R and 4R tauopathies induce differential mitochondrial dysfunction in iPSC-derived astrocytes. (**A**) Schematic representation of the protocols used to isolate BDF-EVs from patients diagnosed with PiD or PSP and non-demented control subjects (**C**) and to treat iPSC-derived astrocytes. (**B**) Micrographs of the mitochondrial system in human iPSC-derived astrocytes expressing the biosensor MitoTimer before (BL) and after (6 h and 24 h) treatment with BDF-LEVs^PiD^ and BDF-LEVs^PSP^. (**C**) Histogram showing mitochondrial redox state changes in iPSC-derived astrocytes 24 h after BDF-LEVs treatment. (**D**) Histogram showing mitochondrial elongation in iPSC-derived astrocytes 24 h after BDF-LEVs treatment. N  =  cultures/cells/mitochondria; ND-LEVs^CFP^: 3/21/801, ND-LEVs^1N3R^: 3/15/958, ND-LEVs^1N4R^: 4/14/789, BFD-LEVs^C^: 3/19/3883, BFD-LEVs^PiD^: 3/21/749, BFD-LEVs^PSP^: 4/17/2555. Two-way matched ANOVA followed by post hoc analyses (Tukey’s test) were used to compare the effects of treatments with those of the other treatments (### *p* < 0.001) and with the nondemented control group (*** *p* < 0.001). Scale bars are 30 µm.

**Table 1 cells-12-00985-t001:** Demographic, biological, and clinical characteristics of the brain sample donors. The characteristics of the donors who provided brain samples used for BDF isolation are listed (*n* = 5 non-demented control subjects, *n* = 5 PSP patients, and *n* = 5 PiD patients). PMD, postmortem delay; NFTs, neurofibrillary tangles; GFTs, glial fibrillary tangles; NA, not applicable. Braak and Thal stages are the two hallmarks of Alzheimer’s disease. These stages are used here to categorize NFT or amyloid β peptide (Aβ) deposition, respectively, and are classified in stages I to VI for NFT [25] and 1 to 5 for Aβ deposits [26]).

Diagnosis	Gender	Death (y)	PMD (h)	Tau Lesions	Braak	Thal	Cause of Death
**Control**	M	78	19	None	0	0	Invasive aspergillosis
	F	82	NA	None	I	1	Pericarditis
	M	23	24	None	0	0	Myocarditis
	M	59	13	None	0	0	Septic shock
	M	41	11	None	0	0	Suffocation
**PSP**	M	74	9	NFT and GFT	0	1	
	M	90	36	NFT and GFT	0	2	
	M	88	3	NFT and GFT	0	4	
	M	69	17	NFT and GFT	0	0	
	F	79	4	NFT and GFT	0	0	
**PiD**	M	57	22	Pick bodies	NA	0	
	M	71	21	Pick bodies	NA	3	
	F	78	11	Pick bodies and NFT	NA	0	
	M	68	15	Pick bodies	NA	0	
	M	68	8	Pick bodies	NA	0	

## Data Availability

The data presented in this study are openly available on http://doi.org/10.5281/zenodo.7759063.

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
