# Peer review of "Tau Transfer via Extracellular Vesicles Disturbs the Astrocytic Mitochondrial System"

_cells, 2023, doi:10.3390/cells12070985_

Round 1
Reviewer 1 Report
The article titled «Accumulation of Tau in Extracellular Vesicles Disturbs the Astrocytic Mitochondrial System » by Perbet et al. show that tau is transferred from neurons to astrocytes via extracellular vesicles (EVs). Namely, the authors demonstrate that both 3R and 4R are transferred via large EVs, but that 3R and 4R isoforms distinctly impact astrocytic mitochondria: the 3R isoform seems to induce mitochondrial fragmentation and oxidation, while the 4R isoform induces mitochondria branching and decreases oxidation. These key findings were recapitulated using EVs isolated from brain-derived fluid from patients with Pick’s disease (with 3R tau isoform) and progressive supranuclear palsy (with 4R tau isoform). The study is well designed, and the manuscript well written. The figures are informative, and the conclusions supported by the results.
I would have minor concerns and comments that the authors may choose to address:
- Table 1: for non-expert readers, please define Braak and Thal stages in the table legend.
- There is only one sample from females per group, is it because males are more affected by PSP and PiD than females ?
- Figure 1A: To avoid viral diffusion from the SD compartment to AX, the volume gradient was inverted (from AX to SD), as seen in the middle panel (2). If the gradient is maintained until DIV12 (right panel, 3), how can the EVs diffuse from neurons to astrocytes ?
- Figure 4B: the radar chart is informative but lack of clarity. Authors should better explain the labels (ratio? events?) in the figure legend. Also, this type of graph does not inform on data variance.
- According to the data, 3R tau is more aggressive than 4R tau with regards to mitochondria dysfunction. 4R tau seems even to “improve” mitochondrial health by increasing fusion and decreasing oxidation. How this finding can be linked to the pathological state in patient? Is PiD more aggressive than PSP? Would there be a link with disease duration / gravity of symptoms?
- Line 499: “4R tau binds to microtubules with a 3-fold higher affinity than (3R?) tau”
Author Response
Comments and Suggestions for Authors
The article titled «Accumulation of Tau in Extracellular Vesicles Disturbs the Astrocytic Mitochondrial System » by Perbet et al. show that tau is transferred from neurons to astrocytes via extracellular vesicles (E.V.s). Namely, the authors demonstrate that both 3R and 4R are transferred via large E.V.s, but that 3R and 4R isoforms distinctly impact astrocytic mitochondria: the 3R isoform seems to induce mitochondrial fragmentation and oxidation, while the 4R isoform induces mitochondria branching and decreases oxidation. These key findings were recapitulated using E.V.s isolated from brain-derived fluid from patients with Pick’s disease (with 3R tau isoform) and progressive supranuclear palsy (with 4R tau isoform). The study is well designed, and the manuscript well written. The figures are informative, and the conclusions supported by the results.
I would have minor concerns and comments that the authors may choose to address:
- Table 1: for non-expert readers, please define Braak and Thal stages in the table legend: The reviewer is right and we have now added definition in the table legend:
Answer1 : We have now completed the table legend:
Table 1- Demographic, biological, and clinical characteristics of the brain sample donors. The characteristics of the donors who provided brain samples used for B.D.F. isolation are listed (n = 5 non-demented control subjects, n = 5 PSP patients, and n = 5 PiD patients). P.M.D., postmortem delay; NFTs, neurofibrillary tangles, G.F.T.s, glial fibrillary tangles; N.A., not applicable. Braak and Thal stages are the two hallmarks of Alzheimer’s disease. These stages are used here to stage NFT or amyloid bï€ peptide (Ab deposition respectively and are classified in stages (I to VI for NFT (Braak et al., 2006) and 1 to 5 for Abï€ deposits (Thal et al., 2002)).
- There is only one sample from females per group, is it because males are more affected by PSP and PiD than females ?
Answer 2: Indeed, gender differences exist in PSP. A retrospective review of medical records of patients diagnosed with PSP over a 21-year period was done in 2002 (Mahale et al., 2002: Gender differences in progressive supranuclear palsy; Acta Neurol Belg. 2022 Apr;122(2):357-362). Data was analyzed from the case records of 334 patients with PSP. 209 patients (62.2%) were male and 125 (37.4%) among the patients were women (male:female ratio = 1.6:1). Regarding Pick disease, this pathology belongs to the Fronto Temporal Disease’s (FTD) group and to be closest to initial neuroanatomical description, we included in our study only Pick bodies without neurofibrillary tangles. This explain why we only 5 cases were available in Lille Neurobank and this may also explain the gender repartition. The same is true for the control group. It should, however, be mentioned that a gender difference has been reported in FTD (not only Pick disease) as described in 2016, with a three-to-4.7-fold greater prevalence in males than in females (Podcasy et al., 2016: Considering sex and gender in Alzheimer disease and other dementias; Dialogues clin Neurosci. 2016 18(4):437-446). An oldier study from 1987 also reported a tendency to higher incidence in males in Pick disease (Heston et al., 1987: Pick's disease. Clinical genetics and natural history; Arch Gen Psychiatry 44:409).
- Figure 1A: To avoid viral diffusion from the S.D. compartment to A.X., the volume gradient was inverted (from A.X. to S.D.), as seen in the middle panel (2). If the gradient is maintained until DIV12 (right panel, 3), how can the E.V.s diffuse from neurons to astrocytes?
Answer 3: Indeed, we maintained the gradient until the end of the experience. It is required to avoid viral diffusion from the somatodendritic compartment to the axonal one so we had no choice. We previously validated these controls (Dujardin et al., 2014). Nevertheless, literature strongly suggest that, at least in a neuron-to-neuron transfer, the synapse is implied. It means that E.V.s are secreted in the axonal compartment from the neurons and directly captured by closed astrocytes. But the reviewer is right, we cannot exclude that part of the E.V.s are moving from the axonal to the somatodentriitc compartment due to the gradient. If it is the case, we probably underestimate the amount of transfer from neurons to astrocytes. But here again, we had no choice because reversing the gradient would induced transport of viral particles from the somatodendritic compartment to the axonal one and then it evaluating tau protein transfer to astrocytes would be impossible.
- Figure 4B: the radar chart is informative but lack of clarity. Authors should better explain the labels (ratio? events?) in the figure legend. Also, this type of graph does not inform on data variance.
Answer 4: We have clarified figure 4B by adding: S.E.M., and graphical elements to understand the measured criteria, and we have also improved the details of the measurement in the methods section (2.12. Analysis of mitochondrial redox state, and dynamics by Mitotimer)
- According to the data, 3R tau is more aggressive than 4R tau with regards to mitochondria dysfunction. 4R tau seems even to “improve” mitochondrial health by increasing fusion and decreasing oxidation. How this finding can be linked to the pathological state in patient? Is PiD more aggressive than PSP? Would there be a link with disease duration / gravity of symptoms?
Answer 5 : Pick’s disease is described as a frontal and a focal lesion which is severe in term of tau lesions and indeed much more aggressive than PSP. The average course of the pathology is between 2 and 5 years. The classical pathology is localized form of cerebral gyral atrophy and this atrophy may be so extreme as to reduce some or most of the gyri (Davis and Robertson, Textbook of neuropathology, Second Edition, 1991, Chapter ‘Degenerative disease of the central nervous system’ pages 918-920). On the contrary, PSP runs a progressive course with death occurring usually within 10 years. Brain weights are usually normal and external inspection unrewarding. There is little evidence of cortical atrophy even so a striatonigra degeneration leading to an atrophy of the caudate and the putamen is observed (Davis and Robertson, Textbook of neuropathology, Second Edition, Chapter ‘Degenerative disease of the central nervous system’ pages 934-936).
We thank the reviewer for this comment and propose to add a comment to the discussion:
‘Treatment with accumulated tau-containing E.V.s originating from neurons with 3R tau accumulation disturbed the astrocytic mitochondrial system and had very damaging effects. ‘These data are in accordance with the severity of Pick disease. Indeed, on contrary to PSP, Pick disease is much more aggressive and damaging with a huge cortical atrophy (Davis and Robertson, Textbook of neuropathology, Second Edition, 1991, Chapter ‘Degenerative disease of the central nervous system’)’
- Line 499: “4R tau binds to microtubules with a 3-fold higher affinity than (3R?) tau”
Answer 6 : This sentence is now modified in the revised version
Reviewer 2 Report
The manuscript addresses the question of the role of extracellular vesicles in the spreading of Tau.
The experimental work is quite good and the authors show convincingly that tau can travel from neurons to astrocytes and that most of the tau accumulation in astrocytes is from tau in EVs.
There are a number of issues that should be addressed before publication.
In figure 1 the difference between Tau accumulation in the neurons versus control looks much greater in 2b than the two fold difference seen in the quantitation in 2c. In Figure 2E the label on the x axis says Tau-V5 OD but the legend states number of tau inclusions. This is not the same thing.
In Figure2D and 2E there is quite a lot of > 150 nm EVs in the "small" EVs and a lot of <150 nm in the "large" EVs. Is there any other parameter used to separate the SEVs and LEV except size?
In figure 4 the authors measure directly the effect of tau on mitochondria of cells in culture. Why are the cells only followed for 24 hours. A longer measure of the effects would be of interest as tauopathies are long term diseases. I am also not sure the heat map in 4c in necessary as the significant changes are also marked in 4b. Also a better explanation of the parameters measured in 4B would be appreciated. Why wasn't the same method used for measuring the effect of brain isolated EVs in 4E, 4F , 4G?
Author Response
Comments and Suggestions for Authors
The manuscript addresses the question of the role of extracellular vesicles in the spreading of Tau.
The experimental work is quite good and the authors show convincingly that tau can travel from neurons to astrocytes and that most of the tau accumulation in astrocytes is from tau in E.V.s.
There are a number of issues that should be addressed before publication.
- In figure 1 the difference between Tau accumulation in the neurons versus control looks much greater in 2b than the two fold difference seen in the quantitation in 2c. In Figure 2E the label on the x axis says Tau-V5 OD but the legend states number of tau inclusions. This is not the same thing.
Answer 1 : Images presented in figure 1B are illustrations to show the existence of a transfer between neurons and astrocytes. These images are selected and then might give the impression of a huge transfer. However, and because these images have been selected, a direct comparison between fig 1B and Fig1C; that is a quantitative assay taken into consideration many cells (5 regions of interest (R.O.I.) in each compartment (somatodendritic compartment, microgrooves, and axonal compartment); is not correct.
- In Figure2D and 2E there is quite a lot of > 150 nm E.V.s in the "small" E.V.s and a lot of <150 nm in the "large" E.V.s. Is there any other parameter used to separate the S.E.V.s and LEV except size?
Answe 2 r: We agree with the reviewer and this is a real concern of the E.V.’s field. The fractions we prepared are enriched in small or in large vesicles but are not pure. Unfortunately, no specific markers have yet been validated by the scientific community, even if researchers are working on and some data are now coming. People are especially using proteomics analyses to try to identified specific markers that should ideally be a transmembrane marker to isolate these vesicles. Unfortunately, this will require an immunoselection and using vesicles coming from an immunoselection in biological assay is very complicated without interferences due to the antibody. We hope that in the future it will be possible to use purer fractions but yet we are only able to prepare enriched fractions.
- In figure 4 the authors measure directly the effect of tau on mitochondria of cells in culture. Why are the cells only followed for 24 hours. A longer measure of the effects would be of interest as tauopathies are long term diseases. I am also not sure the heat map in 4c in necessary as the significant changes are also marked in 4b.
Answer 3: In this study, we had initially decided to follow the cells at 6, 24h, 72h and 7d after treatment. However, as we normalized the different mitochondrial variables to their initial state (BL), the number of usable cells, especially after 24h, decreased significantly (more than 60%). Indeed, many astrocytes migrated a lot in field, proliferated or detached from the plate. The heatmap in 4C (which represents the differences between the 3R and 4R treatments) was intended to simplify the reading of the radar graph. On your advice, we decided to combine this in the same figure by putting the 2 types of statistics and explaining it in the legend.
Also a better explanation of the parameters measured in 4B would be appreciated.
Answer 4 : We have clarified figure 4B by adding: SEM, and graphical elements to understand the measured criteria. We have improved the details of the measurement in the methods section :
Why wasn't the same method used for measuring the effect of brain isolated E.V.s in 4E, 4F , 4G?
Answer 5: If we understand your question correctly, although the methodology is identical, as human astrocytes derived from IPSCs are more challenging to obtain than primary astrocytes, we decided to analyze only the variables observed as significant in the primary culture part at 24H. The detailed and complete effects of EVs (LEV vs SEV) on human astrocytes are under investigation, and we hope they will be the subject of another future publication.
Reviewer 3 Report
RE: Perbet et al.
In this manuscript, the authors studied the transfer of V5-tagged 3R and 4R isoforms of tau in cultures of neurons and astrocytes using a lentivrus expression system, and show that extracellular vesicles (EVs) of neuronal origin enriched with these tau isoforms serve as vehicles for neuron-astrocyte shuttling of tau. Furthermore, the authors also show that exposure to tau containing neuronal EVs altered mitochondrial morphology and redox status, implicating a deleterious effect on astrocytes. The authors also show that EVs isolated from post-mortem human prefrontal cortex with tauopathies Pick disease and PSP also alter mitochondrial morphology and function in human iPSC derived astrocytes. These findings are highly interesting and the experimental design is also impressive, as well as the manuscript is also well written. Knowing the pathological association of glial tau accumulation in human pathologies, which the authors have nicely elaborated, their findings are certainly thought provoking.
I am concerned that the authors may be over-interpreting the data, and require additional control measurements and complementary assays to strengthen their main conclusions. Following are some Major points which the authors need to consider:
Fig2 and 3
The V5-tau ELISA results (Fig. 2G) show that the predominant portion of neuronally secreted tau (both isoforms) is found in the EVs depleted fraction (ND-FFP), followed by large vesicle (ND-LEVs) and the lowest in small vesicles (ND-SEVs). The authors then added 10 microliters of these purified fractions (described in Methods) to cultures of rat hippocampal astrocytes and assessed the abundance of V5-tau in these cultures by V5-ELISA and fluorescence image analyses. Based on the ELISA estimates (2G), the amount of free or EVs-associated tau added onto the cultures in these experiments (Fig 3) would be approximately 25-30 pg (ND-FFP), less than 1 pg (ND-SEVs) and between 1-2 pg (ND-LEVs). However, it is noteworthy that the authors used different EV sample preparation (RIPA suspended for biochemical analyses or phosphate suspended for culture applications). Is the yield in RIPA vs phosphate comparable?
The authors show that after 24 hours (ELISA in Fig. 3C) V5-tau was detected in astrocytes predominantly under conditions of ND-LEVs, while ND-FFP and ND-SEVs show little if any signal. Based on these estimates (2G), on V5-tau abundance in the applied volume of ND-LEVs (approx. 1-2 pg), the levels detected in 3C seem to be unusually high, considering that the fraction was applied to 10x10e5 astrocytes in culture (Methods). Moreover, the authors could elaborate on possible reasons in the context of the literature that despite high abundance of ‘free’ V5-tau in the ND-FFP (both isoforms; Fig 2G), very little is detected in the astrocytes after the addition of this fraction (nearly baseline signal, compare with the sample NI shown in Fig. 3C). See for example:
DOI: 10.1186/s40478-019-0682-x
DOI: 10.3389/fnins.2019.00442
DOI: 10.1084/jem.20172158
DOI: 10.1002/glia.23163
The authors could support their assertion that the ND-LEVs associated V5-tau is preferentially accumulating in recipient cells by, for example, disrupting the EVs with RIPA, applying these fractions to the cells and comparing the levels of accumulation post-exposure (ie, ND-SEVs and ND-LEVs containing equal concentration of V5-tau and compare to ND-FFP fraction normalized in concentration to the others). One would expect that disruption of LEVs with RIPA and then applying onto the cells would also reduce the accumulation.
In 3D, the authors by fluorescence signal intensity analyses show that roughly 1-1.5% V5-tau is detected in the astrocytes under ND-LEVs conditions. The authors should provide additional measurements, such as assess the level of V5-tau in the media samples from these astrocytes cultures (eg, ELISA) since that would reveal how much V5-tau was present with the addition of 10 microliters of fractions (time 0) and at 24 hours.
The use of the work ‘uptake’ is also misleading and authors should revise, it is unclear if the V5 fluorescence signal is within the cytosol or non-specifically in cell membrane and astrocytic processes. The author should perform co-stains with subcellular markers for membrane and ideally 1-2 other mitochondrial markers (see below)
Fig4
Here, the authors assert that the exposure of astrocytic cultures to ND-LEVs associated V5-tau causes alterations in mitochondrial structure and function, with differential effects of 3R or 4R tau isoform. As in my comments above on the concentration estimates, the amount of V5-tau entering the cells is fairly negligible to have profound effects shown (especially within 6 hours), as first it is unclear how much V5-tau is actually detected in cytosol, compared to membrane bound or free in the medium.
A bigger concern is that the authors have relied on a fluorescence microscopy based method (Mitotimer) without providing additional complimentary evidence with other assays of mitochondrial function (eg, morphology, redox status, number). For example, the authors could use some of the mitochondria redox sensitive dyes (Mito-HE, MitoPY1), memebrane potential (JC1) and mitotracker with flow cytometry to complement these data.
The authors should indicate the limitations of the Mitotimer method.
Minor
It is also unclear if V5-tau (both isoform) associates with mitochondria to cause these effects or are independent of such physical association. Based on the differential effects shown for 3R and 4R V5-tau (Fig. 4A), one would expect that mitochondrial targeting in an overexpression system would settle this.
10.1080/21592799.2016.1247939
It is likely that a fraction of V5-tau (3R or 4R) entering the cells interacts with native tau (as the authors elaborate in the discussion) and in such scenario the differential effects are not necessarily due to the delivery of tau through LEVs, but the abundance of tau in the cells. This could be resolved by performing the assays (Fig. 4A-B, along with complementary methods) in astrocytes lacking tau expression (eg, knockout/knock down).
Text
I suggest that the authors should revise the title and replace “Accumulation of tau in extracellular vesicles” to something like “Extracellular vesicles mediated delivery of tau from neurons to astrocytes...
Regarding brain extracts (Fig. 4D-F), the term brain derived fluid (BDF) should be replaced with brain extracted EVs/brain homogenates or something similar (it is a suspension and not a directly collected fluid sample), since BDF can be misleading to connote cerebrospinal fluid.
In the Discussion, ‘We also validated the dependence of astrocytes on neurons for tau spreading’....this assertion is not substantiated and should be removed/revised.
Author Response
In this manuscript, the authors studied the transfer of V5-tagged 3R and 4R isoforms of tau in cultures of neurons and astrocytes using a lentivirus expression system, and show that extracellular vesicles (E.V.s) of neuronal origin enriched with these tau isoforms serve as vehicles for neuron-astrocyte shuttling of tau. Furthermore, the authors also show that exposure to tau containing neuronal E.V.s altered mitochondrial morphology and redox status, implicating a deleterious effect on astrocytes. The authors also show that E.V.s isolated from post-mortem human prefrontal cortex with tauopathies Pick disease and PSP also alter mitochondrial morphology and function in human iPSC derived astrocytes. These findings are highly interesting and the experimental design is also impressive, as well as the manuscript is also well written. Knowing the pathological association of glial tau accumulation in human pathologies, which the authors have nicely elaborated, their findings are certainly thought provoking.
I am concerned that the authors may be over-interpreting the data, and require additional control measurements and complementary assays to strengthen their main conclusions. Following are some
1) Major points which the authors need to consider:
- Fig2 and 3: The V5-tau ELISA results (Fig. 2G) show that the predominant portion of neuronally secreted tau (both isoforms) is found in the E.V.s depleted fraction (ND-FFP), followed by large vesicle (ND-LEVs) and the lowest in small vesicles (ND-SEVs). The authors then added 10 microliters of these purified fractions (described in Methods) to cultures of rat hippocampal astrocytes and assessed the abundance of V5-tau in these cultures by V5-ELISA and fluorescence image analyses. Based on the ELISA estimates (2G), the amount of free or EVs-associated tau added onto the cultures in these experiments (Fig 3) would be approximately 25-30 pg (ND-FFP), less than 1 pg (ND-SEVs) and between 1-2 pg (ND-LEVs). However, it is noteworthy that the authors used different E.V. sample preparation (RIPA suspended for biochemical analyses or phosphate suspended for culture applications). Is the yield in RIPA vs phosphate comparable?
- The authors show that after 24 hours (ELISA in Fig. 3C) V5-tau was detected in astrocytes predominantly under conditions of ND-LEVs, while ND-FFP and ND-SEVs show little if any signal. Based on these estimates (2G), on V5-tau abundance in the applied volume of ND-LEVs (approx. 1-2 pg), the levels detected in 3C seem to be unusually high, considering that the fraction was applied to 10x10e5 astrocytes in culture (Methods).
- In 3D, the authors by fluorescence signal intensity analyses show that roughly 1-1.5% V5-tau is detected in the astrocytes under ND-LEVs conditions. The authors should provide additional measurements, such as assess the level of V5-tau in the media samples from these astrocytes cultures (eg, ELISA) since that would reveal how much V5-tau was present with the addition of 10 microliters of fractions (time 0) and at 24 hours.
Answer 1 : We apologize for the confusion; our explanations were not clear enough.
We determined tau concentration in our E.V.s fractions and F.F.P. at t0 after dilution in astrocytes media. This allowed us not only to compare tau secretion in E.V.s versus F.F.P. (Fig 2) but also to determine tau uptake efficiency in astrocytes.
We have now modified parts of the material and method section:
E.V.s isolation from primary cultures: ‘Media of neurons (10x106; 10 wells) was collected and placed on ice 10 days after LV infection’. ‘ND-LEVs and ND-SEVs were suspended in 100 µl of RIPA buffer (150 mM NaCl, 1% NP40, 0.5% sodium deoxycholate, 0.1% S.D.S., and 50 mM Tris HCl; pH= 8.0) for biochemical assays.’
ELISA: ‘Fractions were obtained after ultracentrifugation of culture medium or B.D.F. as described above and 10 µL were added to astrocytes. Tau levels were then determined using INNOTEST hTau Ag (Fujirebio/Innogenetics, Belgium), which is a sandwich ELISA microplate assay for the quantitative determination of human tau antigen levels in fluids, according to the manufacturer's instructions’.
Rat primary neuron and astrocyte cultures: ‘For the neuron-to-astrocyte tau transfer experiment, 105 astrocytes (1 well) were treated with 10 µl of neuron-derived small E.V.s (ND-SEVs), neuron-derived large E.V. (ND-LEVs) and neuron-derived free protein (ND-FFP) (obtained from 106 neurons, 1 well) at days in vitro 10 (D.I.V. 10).’
To compare tau present in the medium of neurons overexpressing tau and tau transfer to astrocytes we must compare tau quantities and not tau concentration. For the transfer experience, we had 10 µL of vesicles in 3000 µL of medium and quantified tau concentration à this time=t0. As an illustration, in the group of ND-LEV3Rtau, the mean concentration at t0 was 120 pg/mL meaning 360 pg in 3000 µL so a mean uptake efficiency of 1.35 %
To be clearer we have now modified the formula in the material and methods section: ‘Tau uptake efficiency was calculated with the following equation: % of tau uptake= [Quantity of tau in astrocytes x 100] / [Quantity of tau added to astrocytes].’
We also apologize as we inverse the value of 3Rtau and 4Rtau in figure 3D. A non-significant difference is noted between these two groups so this mistake is without consequence in term of results interpretation but we have now corrected that in the revised version.
- Moreover, the authors could elaborate on possible reasons in the context of the literature that despite high abundance of ‘free’ V5-tau in the ND-FFP (both isoforms; Fig 2G), very little is detected in the astrocytes after the addition of this fraction (nearly baseline signal, compare with the sample N.I. shown in Fig. 3C). See for example:DOI: 10.1186/s40478-019-0682-x, DOI: 10.3389/fnins.2019.00442,DOI: 10.1084/jem.20172158, DOI: 10.1002/glia.23163
Answer 2: We thank the reviewer for this suggestion. Indeed, on contrary to other studies we were not able to visualize uptake of free forms of tau. However, the design of these experiments is very different from ours in that we added proteoforms secreted from neurons overexpressin tau (1N3R or 1N4R isoforms) and not recombinant full-length tau (2N4R isoform) (Perea et al.,2019; Piacentini et al., 2017) or pre-formed truncated fibrils (K18P301L, a mutation that is not found in primary tauopathy) (Martini-Stoica et al., 2018). It is now largely described and admitted that many proteoforms of tau are found in the biological fluids (Barthelemy et al., J.A.D. 2016, Sato et al., 2018, Horie et al., 2022) meaning that tau is probably proceed in neurons and secreted in truncated forms. Using free forms of tau coming from neuronal tau secretion allowed us to applied different proteoforms of tau on astrocytes. This might explain why these tau proteoforms are not internalized in astrocytes in our conditions probably because they lost their ability to bind to specific receptors on contrary to a full length-form and as described for neurons (Holmes et al., 2013-Kosik et al., 2020). Extracellular vesicles that contained tau express many proteins at their surface that might interact with specific receptors to support uptake into astrocyte. The receptor/ligand couple(s) remain to be identify and proteomic studies are now ongoing to investigate the vesicular content (Muraoka et al., 2020-Cells).
We then propose to add a comment in the discussion: ‘Contrary to us, some other studies claimed that free tau is uptaken in astrocytes. They applied full-length recombinant proteins (2N4R isoform) (Perea et al.,2019; Piacentini et al., 2017) or preformed fibrils of human tau with P301L mutation not found in primary tauopathies such as PSP or Pick’s disease (K18-P301L) (Martini-Stoica et al., 2018 et al). In our experimental design, we applied neuronal tau secretome. We speculate that this secretome is closer to pathology as many studies now validated the presence of tau peptides in biological fluids and not exclusively the full-length tau (Barthelemy et al., J.A.D. 2016, Sato et al., 2018, Horie et al., 2022).’
- The authors could support their assertion that the ND-LEVs associated V5-tau is preferentially accumulating in recipient cells by, for example, disrupting the E.V.s with RIPA, applying these fractions to the cells and comparing the levels of accumulation post-exposure (ie, ND-SEVs and ND-LEVs containing equal concentration of V5-tau and compare to ND-FFP fraction normalized in concentration to the others). One would expect that disruption of LEVs with RIPA and then applying onto the cells would also reduce the accumulation.
Answer 3: The reviewer is right and this control might be very useful. Disrupting the E.V.s might indeed help us to validate the place of E.V.s in tau transfer to astrocytes. However, it is technically tricky as using RIPA for instance will for sure disrupt the E.V.’s membranes and release its content but, when applied to astrocytes, it will also be damaging for the astrocytes plasma membrane. We already tested it and cells did not survive to RIPA. We should replace RIPA by sonication but the presence of microsomes coming from recircularization of plasma membrane fragments may encapsulate tau or at least modified its uptake. By the past we nevertheless included another control to demonstrate that tau is really inside the vesicles and not stuck at their plasma membrane using proteinase K digestion (Leroux et al., 2020).
- The use of the work ‘uptake’ is also misleading and authors should revise, it is unclear if the V5 fluorescence signal is within the cytosol or non-specifically in cell membrane and astrocytic processes. The author should perform co-stains with subcellular markers for membrane and ideally 1-2 other mitochondrial markers (see below)..
Answer 4: The localization of Tau signal in astrocytes is a fascinating topic, however, it is important to note that extracellular vesicles (EVs) can be internalized by multiple pathways, including clathrin-mediated endocytosis, lipid-raft mediated, caveolin-mediated endocytosis, phagocytosis, or micropinocytosis. These pathways may not be mutually exclusive and can coexist for the internalization of the same set of exosomes. Despite this, little is currently known about how EVs fuse with the plasma membrane of astrocytes and release their contents into the cytosol. In this study, we aimed to demonstrate that the nature of the vesicles can alter the overall response of the astrocyte, rather than investigate the mechanism underlying the uptake of different forms of tau by the astrocyte. As uptake is a dynamic process, addressing this question would require extensive electron microscopy at different time points to confirm the membrane or mitochondrial location of V5 tau and any potential differences between 3R and 4R tau. Additional experiments such as live imaging with TIRFmicroscope (using vesicles stained with PKH26 and mitochondrial labeling with GFP) are currently being considered. This type of analysis is ongoing in our laboratory, and the results will be included in a future publication.
- Fig4: Here, the authors assert that the exposure of astrocytic cultures to ND-LEVs associated V5-tau causes alterations in mitochondrial structure and function, with differential effects of 3R or 4R tau isoform. As in my comments above on the concentration estimates, the amount of V5-tau entering the cells is fairly negligible to have profound effects shown (especially within 6 hours), as first it is unclear how much V5-tau is actually detected in cytosol, compared to membrane bound or free in the medium.
Answer 5: Thank you for your comment. Several studies have demonstrated the remarkable ability of the mitochondrial system of astrocytes to adapt rapidly (<6h) to compensate for damage through numerous mechanisms (https://doi.org/10.3389/fnins.2020.536682).In our study, we found that astrocytes were highly responsive to extracellular vesicles (EVs) containing low amounts of Tau. However, it should be noted that our detection method relied on the presence of the V5 epitope and did not exclude the possibility of other peptides of Tau being present. Under culture conditions, astrocytes typically contain approximately 150 times less tau than neurons, and even small imbalances between the 3R and 4R isoforms of Tau can have rapid consequences on astrocyte reactivity, as recently shown in another study (doi: 10.1172/jci.insight.152012).The kinetics and fate of tau-V5 in astrocytes are also intriguing questions. We currently have limited knowledge about the fate of EVs in astrocytes and their cargo. However, a recent study on HEK cells has shown that EVs can deliver a protein to the cytosol, which subsequently localizes to the mitochondria (https://doi.org/10.1016/j.celrep.2022.110651). It is possible that similar mechanisms could transport Tau to the mitochondria, and this will be the focus of a more in-depth study.
A bigger concern is that the authors have relied on a fluorescence microscopy based method (Mitotimer) without providing additional complimentary evidence with other assays of mitochondrial function (eg, morphology, redox status, number). For example, the authors could use some of the mitochondria redox sensitive dyes (Mito-HE, MitoPY1), membrane potential (JC1) and mitotracker with flow cytometry to complement these data.
Answer 6: Thank you for your comment. Among mitochondrial sensors , MitoTimer has a great advantage that it expresses itself strongly and hardly bleaches, even after repeteted aquisitions. Using the green (488 nm) and red (555 nm) fluorescence ratio permits simultaneous evaluation of individual mitochondria, their morphology analysis, fusion/fission events, and redox state history.This unique property can be used to investigate many scientific questions regarding mitochondria's physiological and pathological roles and is therefore very promising for unveiling the underlying mechanisms of mitochondrial dynamics within many different cell types. This tool alone has already been used in many works to reveal the consequence on the mitochondrial system of different cell types: https://doi.org/10.1042/BCJ20190616 ;https://doi.org/10.4161/auto.26501;doi.10.1074/jbc.M113.530527. In lentiviral construction with detargeting system for neuron (mir124T) and under strong GFAP promoter G1B3, MitoTimer permits to follow the changes in mitochondrial morphology (size, shape, surface area), mobility (speed, displacement), and dynamics (fusioning and fissioning events), as well as the overall mitochondrial turnover rate and redox state. The mitotimer tool is a very complete tool to study the effect of treatment over a long period of time on the same cell. In our hand, the use of classic non integrative mitochondrial probes as JC1 or mito-PY1, although interesting, would not really approach a complete conclusion and would be badly adapted to a long-term imaging of individual astrocytes.
- The authors should indicate the limitations of the Mitotimer method.
Answer 7: The limitation of the mitotimer lies mainly in the complexity of the analyses. Indeed, since it reports on redox state, turnover, dynamics and morphology at the same time, interpretations of changes in some criteria can be tricky. In the past, we have shown that the initial state of the cell can be normalized and that this can greatly improve interpretation and conclusions. We have added more detail in the methods (2.12. Analysis of mitochondrial redox state, morphology and dynamics by Mitotimer)
2) Minor
- It is also unclear if V5-tau (both isoform) associates with mitochondria to cause these effects or are independent of such physical association. Based on the differential effects shown for 3R and 4R V5-tau (Fig. 4A), one would expect that mitochondrial targeting in an overexpression system would settle this.
Answer 8: In our previous study (https://doi.org/10.1038/s41593-020-00728-x), we have already overexpressed Tau 3R-V5 and 4R-V5 in astrocyte cultures. The results obtained with the mitoTimer were very similar to those obtained with 3R and 4R EV treatment, except that for overepxression effects were only visible in astrocytic processes. However, given the pleiotropic nature of Tau it is difficult to envisage that the effects are only direct on mitochondria and we can also envisage that it has different consequences on the regulation of the cytoskeleton as it has been shown in cultured neurons. Although this is very interesting, our study did not aim to show the different mechanisms of these effects.
- It is likely that a fraction of V5-tau (3R or 4R) entering the cells interacts with native tau (as the authors elaborate in the discussion) and in such scenario the differential effects are not necessarily due to the delivery of tau through LEVs, but the abundance of tau in the cells. This could be resolved by performing the assays (Fig. 4A-B, along with complementary methods) in astrocytes lacking tau expression (eg, knockout/knock down).
Answer 9: The basal level of tau in astrocytes is very low (Perea et al., 2019) the way the mitochondrial functions is altered is probably not mediated by the interaction of tau containing in E.V.s and tau in astrocytes. We didn’t address the seeding/nucleation process in the discussion.
3) Text
- I suggest that the authors should revise the title and replace “Accumulation of tau in extracellular vesicles” to something like “Extracellular vesicles mediated delivery of tau from neurons to astrocytes...
Answer 10: We have now modified the title as suggested by the reviewer
- Regarding brain extracts (Fig. 4D-F), the term brain derived fluid (B.D.F.) should be replaced with brain extracted E.V.s/brain homogenates or something similar (it is a suspension and not a directly collected fluid sample), since B.D.F. can be misleading to connote cerebrospinal fluid.
Answer 11 : We agree with the reviewer that the right term is tricky to find. We had huge discussions during the revision of our previous article (Leroux et al., 2022). The reviewer is right; it is a suspension and not a directly collected fluid sample. Because this term is now published we would like to keep it and propose to give more information’s in the material and methods section: ‘BDF-EVs were isolated from human prefrontal cortex tissue as previously described (Polanco et al., 2016; Leroux et al., 2022). This fluid is a suspension obtained by gently papain digestion of brain extract and not a directly collected fluid sample’.
- In the Discussion, ‘We also validated the dependence of astrocytes on neurons for tau spreading’....this assertion is not substantiated and should be removed/revised.
Answer 12: This sentence is now revised in the new version: ‘Narasimham and collaborators also demonstrated that while oligodendroglial tau pathology propagated across the mouse brain in the absence of neuronal tau pathology, astrocytic tau pathology did not. This suggest that pathological forms present in astrocytes are coming from neurons and strengthens the existence of tau spreading between neurons and astrocytes (Narasimham et al., 2020).’
Round 2
Reviewer 3 Report
The authors have satisfactorily addressed the concerns in the revised version. I do not have any further major comments.